# Multi-PGS enhances polygenic prediction by combining 937 polygenic scores

Clara Albiñana [1,2] ✉, Zhihong Zhu[2], Andrew J. Schork[1,3,4], Andrés Ingason[1,3], Hugues Aschard [5], Isabell Brikell[1,6,7], Cynthia M. Bulik [7,8,9], Liselotte V. Petersen[1,2], Esben Agerbo[1,2], Jakob Grove [1,6,10,11], Merete Nordentoft[1,12], David M. Hougaard [1,13], Thomas Werge [1,3,14], Anders D. Børglum [1,6,10], Preben Bo Mortensen [1,2], John J. McGrath [2,15,16], Benjamin M. Neale [17,18], Florian Privé [1,2,20] & Bjarni J. Vilhjálmsson [1,2,11,19,20] ✉

The predictive performance of polygenic scores (PGS) is largely dependent on the number of samples available to train the PGS. Increasing the sample size for a specific phenotype is expensive and takes time, but this sample size can be effectively increased by using genetically correlated phenotypes. We propose a framework to generate multi-PGS from thousands of publicly available genome-wide association studies (GWAS) with no need to individually select the most relevant ones. In this study, the multi-PGS framework increases prediction accuracy over single PGS for all included psychiatric disorders and other available outcomes, with prediction $R^2$ increases of up to 9-fold for attention-deficit/hyperactivity disorder compared to a single PGS. We also generate multi-PGS for phenotypes without an existing GWAS and for case-case predictions. We benchmark the multi-PGS framework against other methods and highlight its potential application to new emerging biobanks.

Although polygenic scores (PGS) have high potential for clinical use[1–3], they are currently underpowered for many applications regarding disease prediction and risk stratification. The predictive performance of PGS is largely determined by four factors: the sample size of the GWAS used for training the score, the proportion of causal variants and the heritability of the phenotype, as well as heterogeneity between GWAS and target samples, including differences in genetic ancestry[4–6]. Increasing the number of samples for a phenotype is costly and takes time, but a possible alternative is to use genetically correlated phenotypes to increase the effective sample size at no cost[1,7–14]. Previously

[1]The Lundbeck Foundation Initiative for Integrative Psychiatric Research, iPSYCH, 8210 Aarhus V, Denmark. [2]National Centre for Register-Based Research, Aarhus University, 8210 Aarhus V, Denmark. [3]Institute of Biological Psychiatry, Mental Health Services, Copenhagen University Hospital, Copenhagen 2100, Denmark. [4]The Translational Genomics Research Institute, Phoenix, AZ, USA. [5]Department of Computational Biology, Institut Pasteur, Université de Paris, 25-28 Rue du Dr Roux, 75015 Paris, France. [6]Department of Biomedicine and Center for Integrative Sequencing, iSEQ, Aarhus University, 8000 Aarhus C, Denmark. [7]Department of Medical Epidemiology and Biostatistics, Karolinska Institute, Stockholm, Sweden. [8]Department of Psychiatry, University of North Carolina at Chapel Hill, Chapel Hill, NC 27514, USA. [9]Department of Nutrition, University of North Carolina at Chapel Hill, Chapel Hill, NC 27514, USA. [10]Center for Genomics and Personalized Medicine, Aarhus University, 8000 Aarhus C, Denmark. [11]Bioinformatics Research Centre, Aarhus University, 8000 Aarhus C, Denmark. [12]Copenhagen Research Centre on Mental Health (CORE), University of Copenhagen, Copenhagen, Denmark. [13]Center for Neonatal Screening, Department for Congenital Disorders, Statens Serum Institut, 2300 Copenhagen S, Denmark. [14]Lundbeck Foundation Centre for GeoGenetics, GLOBE Institute, University of Copenhagen, 1350 Copenhagen K, Denmark. [15]Queensland Centre for Mental Health Research, The Park Centre for Mental Health, Brisbane, QLD 4076, Australia. [16]Queensland Brain Institute, University of Queensland, Brisbane, QLD 4072, Australia. [17]Analytic and Translational Genetics Unit, Massachusetts General Hospital, Boston, MA, USA. [18]Stanley Center for Psychiatric Research, Broad Institute of MIT and Harvard, Cambridge, MA, USA. [19]Novo Nordisk Foundation Center for Genomic Mechanisms, Broad Institute of MIT and Harvard, Cambridge, MA, USA. [20]These authors jointly supervised this work: Florian Privé, Bjarni J. Vilhjálmsson. ✉e-mail: albinanaclara@gmail.com; bjv.ncrr@au.dk

proposed prediction methods using multiple PGS have either required individual-level genotype and phenotype validation data for each PGS in the model[7,11], or the inclusion of multiple PGS for the same GWAS summary statistics file corresponding to different p-value thresholds or proportions of causal variants[15] (i.e. the PGS model hyper-parameters). For the latter, the number of included PGS can become very large (e.g. 10 for each GWAS summary statistic number), which increases the number of validation samples required to fit the model and limits the number of PGS that can be included in practice.

Recent advances in PGS methods allow us to generate scores for a phenotype without requiring validation data to tune the hyper-parameters[16–20]. This development has two major implications in the context of prediction using multiple PGS. First, individual-level genotype validation data for each of the correlated phenotypes included in the prediction model is no longer necessary because selecting the best-performing hyper-parameters is no longer needed. Second, PGS for any genetically correlated phenotype, even those not available in the target data, can now be included easily in the same prediction model, which significantly expands the set of phenotypes one can study. Therefore, as only one PGS per phenotype needs to be included in the prediction model, the only practical limitation is effectively the number of individual-level samples available for the desired pheno-type, which is constantly growing for biobank data.

In this work, we propose and evaluate a multi-PGS framework that leverages these key PGS developments to construct more powerful and generalisable prediction models. These multi-PGS models can be trained on thousands of different PGS such as for health outcomes, body measurements, and behavioral phenotypes which are not necessarily genetically correlated with the outcome. Multiple PGS and covariates can be combined using either a linear model (lasso pena-lized regression) or a nonlinear model (XGBoost[21]) into a multi-PGS model. This model is then evaluated in an independent dataset in terms of the prediction accuracy of the multi-PGS. We apply our multi-PGS framework to the Lundbeck Foundation Initiative for Integrative Psychiatric Research (iPSYCH)[22,23], one of the largest datasets on the genetics of major psychiatric disorders. These disorders are geneti-cally correlated with many other psychiatric and neurological dis-orders as well as other behavioral phenotypes[24,25], which are precisely the circumstances under which the proposed multi-PGS might boost the polygenic prediction accuracy. We benchmark the multi-PGS against each phenotype's respective single PGS prediction and com-pare it with an existing PGS method that meta-analyzes multiple PGSs using GWAS summary statistics, wMT-SBLUP[8]. Although the iPSYCH cohort has been designed around psychiatric disorders, the study individuals can be linked to the National Danish Registers[22,23], making it possible to generate multi-PGS for any phenotype captured in these registers. We demonstrate that multi-PGS improves prediction accu-racy results for a range of different diseases, subtypes and phenotypes for which no GWAS summary statistics currently exist (e.g., birth measurements and case-case classification). Our goal is to showcase our multi-PGS framework and its potential advantage to be applied to new emerging biobank data.

## Results

### Overview of method

Here we summarize the framework used for generating the proposed multi-PGS. This framework consists of three steps (Fig. 1). In Step 1 "Build PGS Library", a large, agnostic library of PGS is generated by running LDpred2-auto[16] on publicly available GWAS summary sta-tistics (GWAS Catalog[26], GWAS ATLAS[27], PGC[28] etc.). In Step 2 "Train Multi-PGS Models", the PGS library is standardized (i.e., mean 0 and variance 1) and used to develop multi-PGS prediction models for a target outcome using both a linear model (lasso penalized regres-sion) and a non-linear model (boosted gradient trees, XGBoost). These models include sex, age and 20 first PCs as covariates for training. Finally in Step 3 "Evaluate models", the multi-PGS models are projected into the test data, evaluated in terms of prediction accuracy and benchmarked against single PGS and another

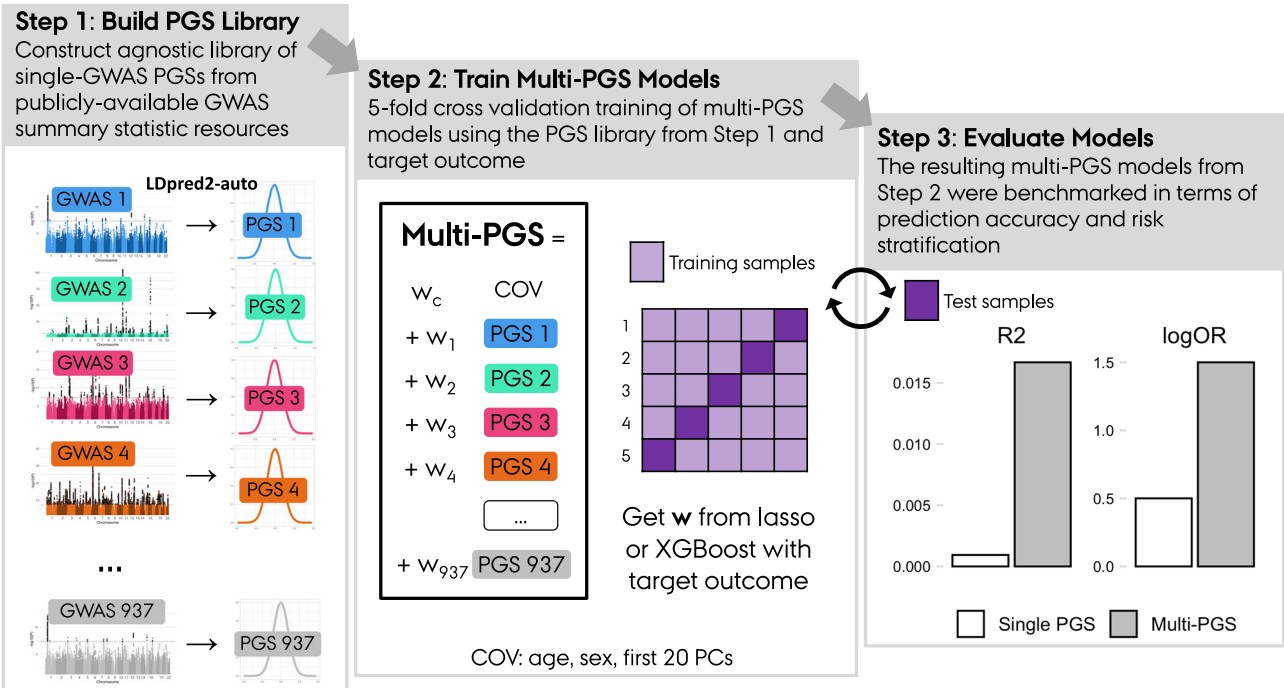

**Fig. 1 | Overview of the multi-PGS framework.** The framework consists of 3 sequential steps: Step 1) Build PGS Library. Construct an agnostic library of single-GWAS PGSs from publicly-available GWAS summary statistic resources. Step 2) Train Multi-PGS Models. Fivefold cross validation training of multi-PGS models using the PGS library from Step 1 and target outcome. Step 3) The resulting multi-PGS models from Step 2 were benchmarked in terms of prediction accuracy and risk stratification.

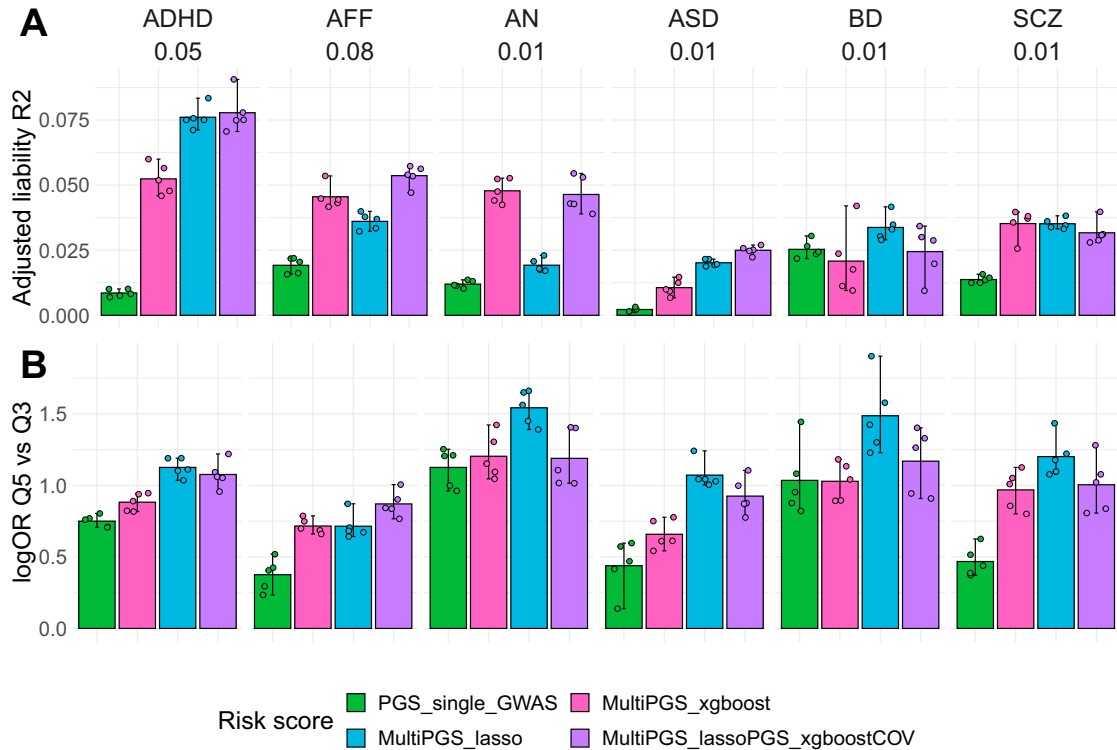

**Fig. 2 | Performance of the different risk scores including covariates.** Comparison between the per-disorder attention-deficit/hyperactivity disorder (ADHD), affective disorder (AFF), anorexia nervosa (AN), autism spectrum disorder (ASD), bipolar disorder (BD) and schizophrenia (SCZ) single GWAS PGS (specific details on SD2) and the multi-PGS trained with 937 PGS in terms of **A** liability adjusted R2 and **B** log odds ratios of the top risk score quintile compared to the middle risk score quintiles. All models included sex, age and first 20 PCs as covariates for training and calculating the risk score on the test set in a fivefold cross-validation scheme. The MultiPGS_lasso and MultiPGS_xgboost were trained with lasso regression and XGBoost respectively, using the 937 PGS and the covariates as explanatory variables. The MultiPGS_lassoPGS_xgboostCOV was generated with lasso regression, combining the 937 PGS and the predicted values of an XGBoost model that included only the covariates. 95% confidence intervals were calculated from 10,000 bootstrap samples of the mean adjusted R2or logOR, where the adjusted $R^2$ was the variance explained by the full model after accounting for the variance explained by a logistic regression covariates-only model as R2adjusted = (R2full - R2cov)/(1 − R2cov). Prevalences used for the liability are shown beneath each disorder label and case-control ratios are available on SD2. All association logOR for all quintiles are available in SF6.

multivariate PGS method (wMT-SBLUP[8]). We used 5-fold cross-validation to alternate between Step 2 and Step 3 to get out-of-sample prediction accuracy estimates.

Using the proposed multi-PGS framework, we generated a library of 937 PGS models (described in detail in Supplementary Methods) and projected it into the genotypes of individuals in iPSYCH. We then trained multi-PGS models for 6 major psychiatric disorders: attention-deficit/hyperactivity disorder (ADHD), affective disorder (AFF), anorexia nervosa (AN), autism spectrum disorder (ASD), bipolar disorder (BD) and schizophrenia (SCZ). We focus the first part of the results section on these 6 psychiatric disorders and extend the multi-PGS application to other 62 ICD10 code disease definitions, continuous phenotypes and case-case classification in the last result section.

### Linear and non-linear combinations of PGS give comparable prediction results

We first studied risk prediction models that combine covariates (sex, age and first 20 PCs) and the 937 PGS using linear models (lasso penalized regression; multiPGS_lasso) and non-linear models (boosted gradient trees: multiPGS_XGBoost) to predict 6 major psychiatric disorders: ADHD, AFF, AN, ASD, BD and SCZ. We used a model including a single PGS for the largest available GWAS for each psychiatric disorder as the standard reference (ST3). In terms of variance explained, the multi-PGS increased the mean $R^2$ 4-fold on average over the single GWAS PGS for all disorders, with up to 9-fold improvement for ADHD and ASD (Fig. 2A), from 0.8 to 7.5 and from 0.2 to 2, respectively. Compared to the middle risk score quintiles, the top multi-PGS quintile

generally increased the log odds ratio over the single GWAS PGS (Fig. 2B).

In terms of which multi-PGS performed better at combining the variables, the results appear to be disorder specific. For ADHD, ASD and BD the lasso multi-PGS increased the mean prediction R2 over the XGBoost multi-PGS. On the contrary, for AFF and AN the XGBoost multi-PGS increased the mean prediction $R^2$ over lasso multi-PGS. For SCZ, there was no difference in variance explained by the two PGS.

We further investigated if the increase in prediction of the XGBoost multi-PGS over lasso multi-PGS was driven by the nonlinear combination of covariates alone, and if it was independent from the PGS combination. In practice, we obtained an XGBoost risk score for the covariates and fitted it as an additional variable in the lasso model, together with the 937 PGS. The mean variance explained by this mixed multi-PGS was comparable to the lasso multi-PGS for ADHD, ASD, BD and SCZ, while it was comparable to the mean variance explained by the XGBoost multi-PGS for AFF and AN. This demonstrated that the non-linear combination was only beneficial at the covariate level and not at the PGS-level.

Less pronounced differences were observed in terms of the mean area under the curve (AUC) prediction (SF5), indicating that both models are similar in terms of classification. In terms of quintile odds ratio, the lasso multi-PGS was generally the best at separating the top 20% to the middle quintile, even for the models where the maximum mean variance was explained by the XGBoost multi-PGS (Fig. 2B). Since the two multi-PGS provided relatively similar results, we continued further analyses on the psychiatric

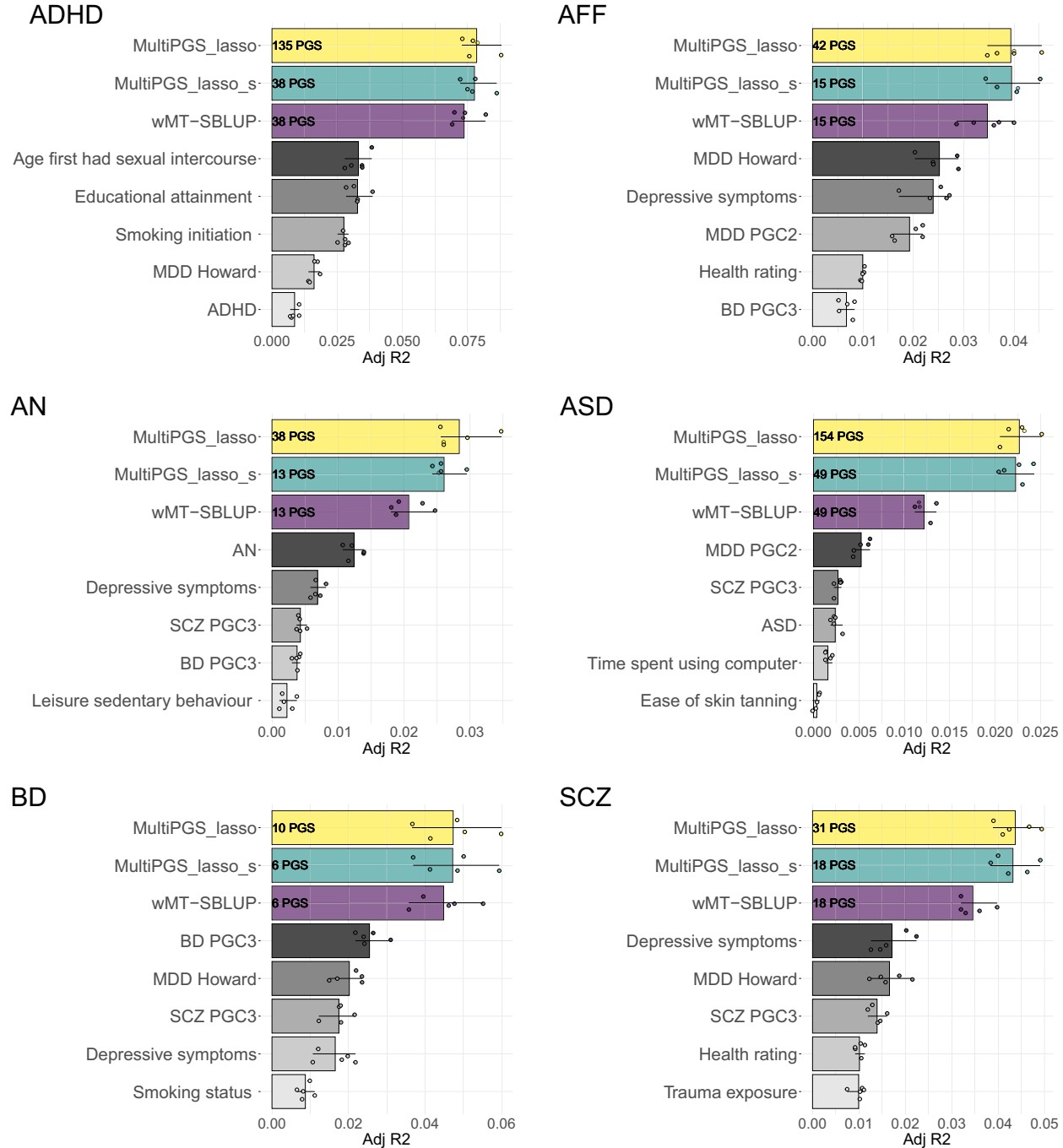

**Fig. 3 | Comparison between single-phenotype and multi-phenotype PGS (multi-PGS and wMT-SBLUP).** Mean liability adjusted R2 estimates between attention-deficit/hyperactivity disorder (ADHD), affective disorder (AFF), anorexia nervosa (AN), autism spectrum disorder (ASD), bipolar disorder (BD) and schizophrenia (SCZ) and multi-phenotype predictors (colored bars, multiPGS_lasso, multiPGS_lasso_s, wMT-SBLUP) or single-phenotype PGS (grayscale bars, single LDpred2-auto PGS). The 5 single-phenotype PGSs shown were selected based on the top ranking absolute lasso weights. The adjusted $R^2$ estimates are the mean of the fivefold cross-validation training-testing subsets. CI were calculated from 10k bootstrap samples of the mean. The numbers inside each multi-phenotype predictor correspond to the number PGS included in each model. Both the simplified multi-PGS (multiPGS_lasso_s) and wMT-SBLUP predictors were calculated by keeping the top PGS with an absolute lasso weights >0.01 from the full multi-PGS, including the top 5 shown in the figure.

disorders considering only the lasso multi-PGS, as weights from linear models are more interpretable.

**Comparison between single PGS and multi-PGS predictors**

Next, we investigated which of the 937 PGS in the multi-PGS model were the ones contributing the most to increasing prediction accuracy. First, we identified the number of non-zero PGS selected by the lasso model, which ranged from 10 to 154 for the 6 psychiatric disorders (Fig. 3, non-zero PGS number inside the multi-PGS_lasso bar plot). All non-zero mean weighted PGS are available at SF7-SF12. While ADHD and ASD had over 100 PGS included, BD had only 10. This could express the larger amount of genetically-correlated phenotypes in the PGS library with ADHD and ASD compared to BD, but could also be a reflection of the training sample size. The larger the training sample

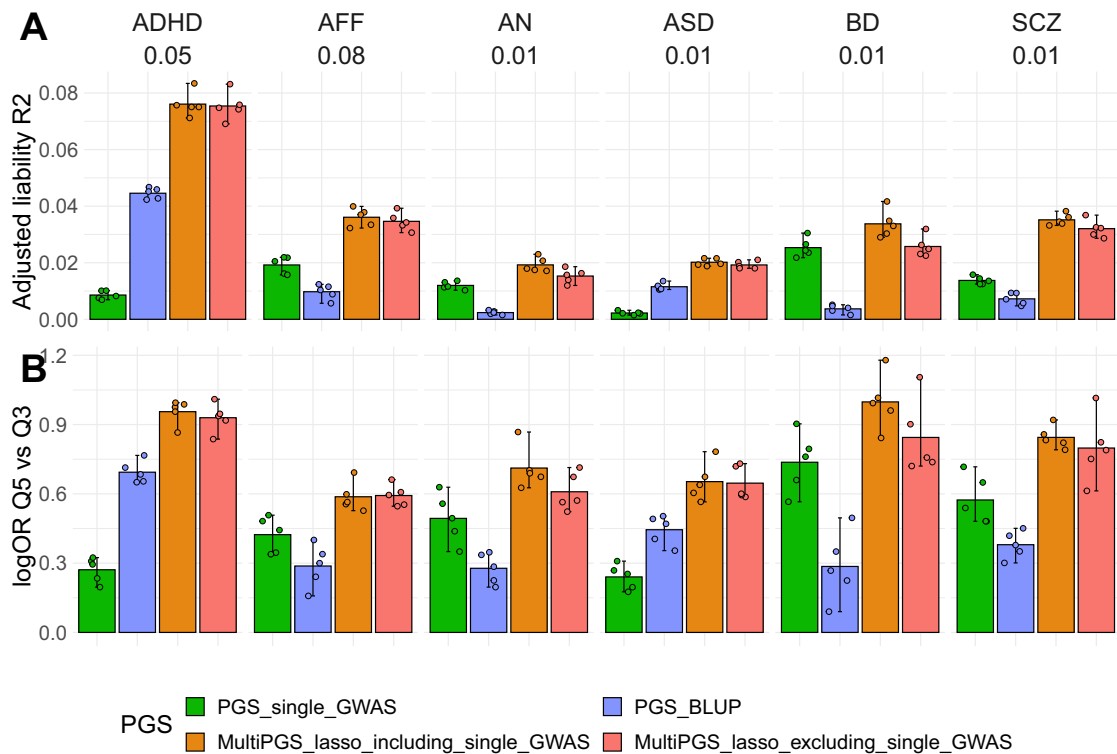

**Fig. 4 | Performance of the PGS trained with different data.** Comparison between the per-disorder attention-deficit/hyperactivity disorder (ADHD), affective disorder (AFF), anorexia nervosa (AN), autism spectrum disorder (ASD), bipolar disorder (BD) and schizophrenia (SCZ) single GWAS PGS (PGS_single_GWAS) (Details on SD2), the per-disorder BLUP PGS and the multi-PGS in terms of **A** liability adjusted $R^2$ and **B** log odd ratios of the top quintile compared to the middle quintile. The multiPGS_lasso_excluding_single_GWAS represents the PGS where the specific single GWAS PGS was removed from the set of 937 PGS. All models were adjusted for sex, age and first 20 PCs. The adjusted liability $R^2$ shows the mean of the fivefold cross-validation training-testing subsets. CI were calculated from 10k bootstrap samples of the mean adjusted $R^2$ or logOR, where the adjusted $R^2$ was the variance explained by the full model after accounting for the variance explained by a logistic-regression covariates-only model as R2_adjusted = (R2_full − R2_cov)/(1 − R2_cov). Prevalences used for the liability are shown beneath each disorder label and case-control ratios are available on SD2. All association OR for all quintiles are available in SF14.

size, the larger the power to identify genetically correlated phenotypes in the PGS library through lasso regression. However, weights were generally very small, with few PGS with an absolute lasso weight larger than 0.01 (Fig. 3, number of PGS with absolute weight larger than 0.01 inside the multi-PGS_lasso_s and wMT-SBLUP bar plots). Among the top 5-weighted PGS for each psychiatric disorder, we always identified the PGS from the PGC GWAS excluding iPSYCH samples for all disorders.

For affective disorder, three depression PGS were selected, two from the PGC[29,30] (tagged MDD-PGC2 and MDD Howard) and one from a UK Biobank GWAS on depressive symptoms[31] (tagged Depressive symptoms). These results confirm that non-overlapping signals from multiple GWAS of similar phenotypes can be combined to increase prediction accuracy. Interestingly, the PGS from the PGC GWAS excluding iPSYCH samples ranked 5th for ADHD. This study had only 4225 cases, similarly to the ASD study, with 5305 cases. The iPSYCH individuals were excluded from the PGC ADHD and ASD studies.

We compared the prediction accuracy of a multi-PGS with all non-zero weights to a simplified lasso multi-PGS that included only PGS with an absolute lasso weight larger than 0.01, multi-PGS_lasso_s (Fig. 3, number of included PGS in the figure), showing very similar mean $R^2$. Prediction estimates for the top 5 PGSs in each model ranked by their lasso weights are also shown in Fig. 3, with the top PGS generally contributing to half of the prediction accuracy of the multi-PGS.

By using the lasso regression as a feature selection algorithm (only selecting the ones with large weights), it was feasible to compare the prediction accuracy to another prediction method using multiple

PGSs, wMT-SBLUP[8], as this method estimates weights for all PGS in the prediction. The PGS weights in wMT-SBLUP are calculated as a function of the GWAS summary statistics' sample size, SNP-heritability and genetic correlation (SF12 contains an overview of these parameters for the top 10 PGSs in each model). The simplified multi-PGS showed consistently higher mean $R^2$ than wMT-SBLUP, even though they both contained the same number of PGS.

**Combining hundreds of external PGS increases prediction over training only on the individual-level data**

One of the potential issues of comparing single-PGS to multi-PGS methods is the use of individual-level data for training in multi-PGS, given that for some analyzed disorders the number of cases is larger in iPSYCH than in the rest of the PGC cohorts (ST3). Here, we compared the prediction accuracy of single-PGS (based on GWAS summary statistics) and multi-PGS (based on both GWAS summary statistics and individual-level data) to a best linear unbiased predictor (BLUP) PGS (based on individual-level data). We used fivefold cross validation for deriving both the multi-PGS and BLUP PGS, so that the reported adjusted $R^2$ are out-of-sample estimates. The resulting comparison showed that the prediction accuracy of multi-PGS outperformed both single-PGS and BLUP PGS predictions, indicating that the accuracy gained from multi-PGS was not only based on sample size but on the use of a large and diverse set of 937 PGS (Fig. 4A). As shown previously[32], the BLUP PGS vs. single GWAS PGS varied in the relative proportion of variance explained according to the psychiatric disorder, as they are largely dependent on the training sample sizes and genetic correlation. We observed these large differences also in terms

of log OR of the PGS quintiles separating the top to the middle quintile of the sample (Fig. 4B).

Next, we further explored the capacity of our multi-PGS to predict outcomes for which there are no available external GWAS summary statistics. This question is inspired by a scenario where the studied outcome could benefit from PGS analyses, but there is still no GWAS for that outcome. We tested this by simulating a scenario where the six analyzed psychiatric disorders did not have external PGC GWAS. In practice this meant removing each time the disorder's PGS from the PGS library, resulting in a PGS library of 936 scores. For affective disorder (AFF), we removed the two depression PGS from the library. The difference in mean prediction R2 was not significantly different between both multi-PGS (using either 937 or 936 PGS) for any psychiatric disorders (Fig. 4A). Moreover, the multi-PGS that did not contain the disorder's external PGS was not significantly worse at separating the risk quintiles in terms of log OR (Fig. 4B). These results indicate that it is possible to generate multi-PGS for a specific outcome without the need of a GWAS for that same phenotype, given that the PGS library contains scores for genetically correlated outcomes.

### Generating multi-PGS from register-based phenotypes

Finally, we extended the multi-PGS results to other phenotypes defined in the Danish National Registers in the overlapping samples with iPSYCH to showcase its potential to generate polygenic scores in a biobank framework. First, we selected 62 ICD10 codes from the Danish Psychiatric Central Research Register[33] with at least 500 diagnosed cases and compared the prediction performance of the multi-PGS lasso to the multi-PGS XGBoost. Similarly to the results for the main 6 psychiatric disorders, the mean prediction R2 for both multi-PGS was on average the same, but ultimately depended on the disorder (SF13). We emphasize that both these phenotypes all nested within the psychiatric disorder cases and population cohort of iPSYCH case-cohort design (ST4).

To expand the comparison to external data, we selected 15 phenotypes from four different categories; (a) other ICD10 codes with available GWAS summary statistics, (b) other ICD10 codes with not known available GWAS summary statistics / disorder sub-phenotypes, (c) continuous phenotypes from the Medical Birth Register[34] and (d) case–case predictions. For the last category, we explored two example pairs of disorders with a high degree of comorbidity. First, we excluded the cases with both disorders and re-codified each single disorder case as 0 or 1.

In this case we compared the prediction accuracy of the multi-PGS against a single-PGS, the top weighted PGS as outputted from the multi-PGS lasso (ST5).

The results varied greatly for each phenotype, but there were large increases in prediction accuracy of the multi-PGS for most phenotypes (Fig. 5). The multi-PGS for the disorder pair ASD/ADHD showed the largest prediction R2 of all examined multi-PGS, explaining 12% of the variation. On the other side of the spectrum, the multi-PGS for outcomes such as APGAR score, gestational age or BD/MDD showed null R2. Note that the PGS labeled MDD Howard[30] is selected as the top lasso weighted PGS for multiple outcomes (Fig. 5). This can be a reflection of the sampling strategy in iPSYCH, where individuals diagnosed with ADHD, AN, AFF, ASD, SCZ and BD were over-sampled compared to the population. The secondary phenotypes' cases shown in Fig. 5 have a large overlap with affective disorder diagnoses (ST4).

## Discussion

Here we have proposed a multi-PGS framework derived from nearly one thousand GWAS summary statistics for different phenotypes, and showed that it increased the accuracy of PGS for psychiatric disorders. Multi-PGS explained a larger proportion of the SNP-heritability (a fourfold increase on average for the main psychiatric disorders over a PGS trained on the target outcome only) and stratified the population

in more distinct risk groups than single-phenotype PGS. The disorder with the largest increase in prediction accuracy from multi-PGS was ADHD, with a ninefold increased prediction R2 over the single PGS (corresponding to ~40% of the SNP $h^{2[35]}$). This is mostly due to the inclusion of various behavioral PGS in our library (educational attainment, smoking status, etc.) with very large sample sizes and large genetic correlation with ADHD.

While the compared *linear* multi-PGS (lasso) and *non-linear* multi-PGS (XGBoost) resulted in similar prediction accuracies, the improved performance of the non-linear models for affective disorder and anorexia nervosa was due to the presence of non-linear covariate interactions between sex and age, as previously reported[36]. We therefore focused the analyses on the lasso multi-PGS, as those results have easier interpretability.

We benchmarked our multi-PGS prediction results against a PGS trained on external GWAS summary statistics for each of the six major psychiatric disorders respectively. The multi-PGS always resulted in a more accurate prediction, even when the phenotype itself was not included in the PGS library. When benchmarked against wMT-SBLUP[8], another PGS method trained on GWAS summary statistics for multiple phenotypes, the multi-PGS also resulted in a more accurate prediction. The improvement in prediction accuracy is likely due to the fact that while wMT-SBLUP bases its weights on the genetic correlation to the external GWAS summary statistics for each phenotype, multi-PGS is trained directly on the individual-level samples, allowing multi-PGS to better tailor the weights to the cohort. Interestingly, the multi-PGS still obtained much more accurate predictions by combining hundreds of external PGS. This result suggests that as both the number of genome-wide association studies and their sample size grows, training multi-PGS will become more feasible.

Finally, we showcased how multi-PGS can be used to train predictors for any phenotype of interest, as long as one has *sufficient* individual-level genetic data available with the phenotype of interest. The multiPGS predictors do not require PGS for the target phenotype of interest to be available in the PGS library used. This application is particularly interesting for sub-phenotype analyses within diseases, where GWAS summary statistics are not generally available for the sub-phenotypes. We demonstrated in practice how these multi-PGS could be generated for various psychiatric sub-diagnoses e.g. different ICD10 subcodes within Autism Spectrum Disorder (ICD10 F8). Similar multi-PGS method could also be applied to other sub-phenotypes of psychiatric disorders like psychosis within bipolar disorder, as defined in Hasseris et al.[37]. Another exciting application we explored is the case-case prediction, where multi-PGS models can be trained for highly comorbid disorders. In this last category, we highlight the relatively high prediction accuracy of our predictor of ADHD cases from a pool of ADHD-ASD cases.

This study and the multi-PGS approach has several limitations. First, as we performed a fivefold cross-validation in the iPSYCH (individual-level) data when training and testing the multi-PGS, it is possible that some of the prediction accuracy gain is due to overfitting. However, the multi-PGS also resulted in multiple null predictions despite having large training sample sizes (APGAR score and gestational age) suggesting that overfitting is small. To further avoid overfitting we restricted the analyses to a set of unrelated individuals of European ancestry to control for population structure. Second, controlling for sample overlap between external and internal data, which can lead to overfitting[38], becomes both difficult and important when considering thousands of GWAS summary statistics. We addressed this manually by checking the GWAS summary statistics, but an automatization of this step (e.g. using bivariate LDSC[39]) could help streamline this procedure. Third, the resulting multi-PGS is potentially predicting a subset of individuals in the case group enriched or comorbid with a phenotype with large genetic correlation to the PGS phenotypes, but not the disorder itself. Therefore, although multi-PGS can improve prediction

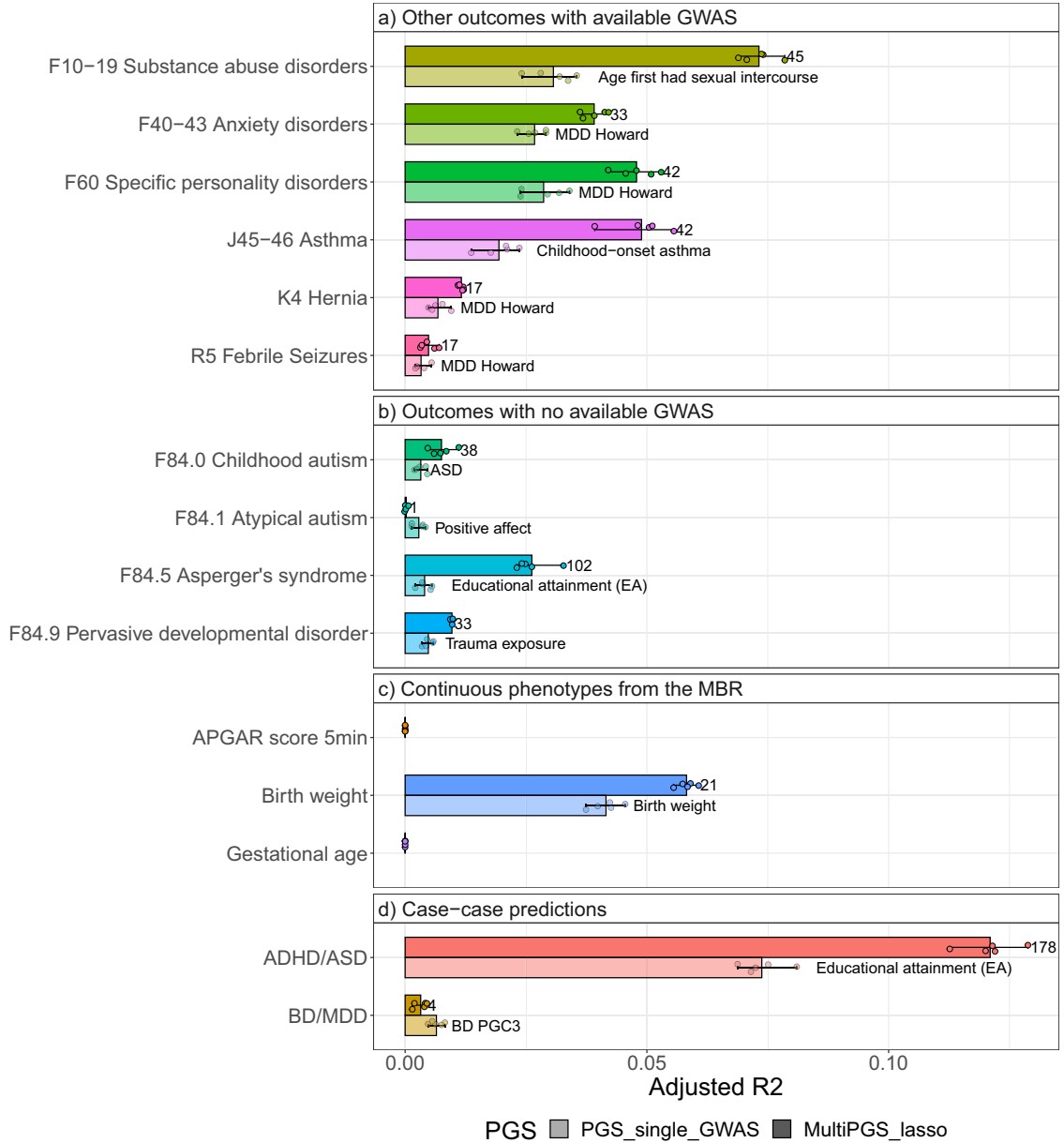

**Fig. 5 | Examples of the prediction accuracy of multi-PGS vs. top predictive single-GWAS-PGS on register-based phenotypes.** Comparison between a per-phenotype single GWAS PGS (the top-ranked PGS with largest weight from the lasso multi-PGS model on each outcome, details on SD4) and the multi-PGS trained with 937 PGS in terms of adjusted $R^2$. The set of outcomes includes **a** other outcomes with available GWAS, **b** outcomes with no available GWAS, **c** continuous phenotypes from the MBR and **d** Case−case predictions. All models included sex, age and first 20 PCs for training the different PGS weights and calculating the risk score on the test set in a fivefold cross-validation scheme. CI were calculated from 10,000 bootstrap samples of the mean adjusted $R^2$, where the adjusted $R^2$ was the variance explained by the full model after accounting for the variance explained by a logistic regression covariates-only model as R2adjusted = (R2full − R2cov)/(1 − R2cov). The number next to the multiPGS bar indicates the number of non-zero lasso mean weights for the 5 cross-validation subsets.

accuracy, it should not be used to estimate genetic correlations or study genetic overlap. Fourth, we have not explored whether including a PRS trained on individual-level data could improve the prediction further, as suggested by a previous study[32]. Fifth, we have not explored how generalisable the multi-PGS are across different ancestry groups, but we expect the $R^2$ prediction accuracy to decay with genetic distance between training and testing, as previously shown for PGS[6,40]. However, as more individuals of non-European ancestry are included in GWAS, the multi-PGS based on the resulting summary statistics may be able to improve cross-ancestry prediction.

The same multi-PGS framework could be applied to other types of biological data summary statistics, like GWAS for brain or cardiac images[41,42], gene expression[43], and/or protein levels[44]. Combining

different types of data into a multi-PGS could potentially improve or quantify the importance for prediction of the different data types. The framework can also make use of published PGS variant weights (e.g. from the PGS catalog[45]) instead of deriving the PGS from the GWAS summary statistics.

In this study, by leveraging nearly a thousand external PGS, we show how we can increase the polygenic prediction further without the need to genotype more individuals of a specific phenotype. We think this multi-PGS framework has a lot of potential for new emerging biobanks or register-based genetic cohorts to generate PGS for every available phecode or defined phenotype in their system, both because of the ever growing set of publicly available GWAS summary statistics and, as they are new, these biobanks do

not have the issue of sample overlap with the external GWAS summary statistics used.

## Methods

The study was approved by the local scientific ethics committees and institutional review boards. The iPSYCH study was approved by the Scientific Ethics Committee in the Central Denmark Region (case no. 1-10-72-287-12) and the Danish Data Protection Agency. In accordance with Danish legislation, the Danish Scientific Ethics Committee has, for this study, waived the need for specific informed consent in biomedical research based on existing biobanks.

### PGS library construction

A detailed description of the PGS library construction and file filtering process is provided as Supplementary Methods and all code used will be available. A number of resources were used to obtain an initial list of GWAS summary statistics for generating the PGS library. The majority of the GWAS summary statistics were downloaded from publicly available databases (GWAS Catalog[26], GWAS Atlas[27], the Psychiatric Genomics Consortium Website (https://www.med.unc.edu/pgc). For the specific PGC GWAS summary statistics where iPSYCH was used in the discovery dataset, we used in-house GWAS results where these samples were excluded from the calculation. The GWAS files were specifically selected to be based on European ancestry individuals, to not be overly redundant (only the latest GWAS for each same phenotype) and to not contain iPSYCH samples. From an original list of 6206 files (ST1), this filtering resulted in 1377 files to download.

We developed a pipeline for downloading, parsing, reformatting and doing a quality control filter on the list of files. We created a GWAS summary statistic column-name library (SF1) to ease file parsing, and after re-formatting and removing corrupted files, this step resulted in 1005 files. We restricted the number of SNPs to the overlap of the iPSYCH imputed variants with the HapMap3 variants and the LD reference provided by LDpred2, resulting in a maximum of 1,053,299 SNPs per GWAS summary statistics file. Filtering SNPs with a large discrepancy in standard deviation between the genotyped/imputed data and the GWAS summary statistics can increase the robustness and prediction accuracy of PGS[46]. For each file, we created a QC plot for visual inspection of the QC SNP filtering (Example in SF2). The set of 952 GWAS summary statistics that passed QC and kept over 200,000 SNPs were used to derive PGS with LDpred2-auto[16].

Polygenic score weights were derived using LDpred2-auto, a method within the LDpred2 framework[16] that does not require a validation dataset to fit the hyperparameters (SNP-h2; SNP-based heritability estimate and p; proportion of causal SNPs), but these are fitted as part of the Gibbs sampler instead. We used the provided European-ancestry independent LD blocks as reference panel[46]. For each GWAS summary statistics file, LDpred2-auto was run with 30 Gibbs sampler chains, 800 burn-in iterations and 400 iterations. The SNP-h2 initial value was set to the LD score regression estimate[47] from the GWAS summary statistics after. Each of the chains was initialized with a different prior for the proportion of causal variants: [1e-4, 0.9] in log scale (example plot for a chain in SF3). Chains were filtered according to the recommendation in the LDpred2 tutorial, and effect sizes of chains kept were averaged. After running LDpred2-auto on the QC'd file set and post-processing, we were left with 937 PGS. This constitutes the final PGS library and all information on its GWAS summary statistics meta-data, number of SNPs per file on each step, number of chains on the final PGS and estimates of SNP-h2 and p can be found in SD1. A plot comparing the LDSC and LDpred2-auto SNP-h2 estimates can be found at SF4.

### iPSYCH data

**Genotypes and imputation.** The iPSYCH 2015 case-cohort sample is a genotyped dataset from neonatal dried blood spots (DBS) nested within the entire Danish population born between 1981 and 2008, including 1,657,449 persons[22,23]. After genotyping and sample quality control, it includes 92,765 individuals diagnosed with a major psychiatric disorder i.e. attention-deficit/hyperactivity disorder (ADHD), affective disorder (AFF), autism (ASD), schizophrenia (SCZ) and bipolar disorder (BD). We also included the anorexia nervosa (AN; ANGI-DK) samples from the Anorexia Nervosa Genetics Initiative (ANGI)[48], as they were samples within the same framework as iPSYCH 2015. The dataset also includes 42,912 individuals randomly sampled from the same birth cohort, making it representative of the general Danish population. The genotype data was imputed using the Haplotype Reference Consortium (HRC)[49] as the reference panel and following the RICOPILI pipeline[50]. After removing SNPs with minor allele frequency (MAF) < 0.01 and Hardy-Weinberg p value < 10-6, we restricted to the HapMap3 variants in the LDpred2 LD reference panel, resulting in 1,053,299 SNPs.

**Principal components & relatedness.** We performed principal component analysis (PCA) following Privé et al.[51] and obtained 20 PCs. The process has already been described in Albiñana et al.[32]. Using the set of 20 PCs, we defined genetically homogeneous individuals as having <4.5 log distance units to the multidimensional center of the 20 PCs (calculated using the function dist_ogk from the R package bigutilsr[51,52]). We also computed the KING-relatedness robust coefficient of the sample and excluded the second of each pair with >3rd degree relatedness. We identified a set of 108,031 unrelated genetically homogeneous individuals (Danish-European ancestry), which we used for all subsequent analyses.

**List of phenotypes and ICD10 codes.** We used phenotypes from the Danish Psychiatric Central Research Register[33] with register data available until December 2016. Except for the category of all psychiatric disorders (ICD10, F-chapter), all categories of diagnosis are given a variable name starting with a letter and followed by four digits. The first letter is the chapter of disease in the ICD10 system and the first number is the ICD10 diagnosis. The other 3 numbers are not informative of ICD10 diagnosis. We also used 3 continuous phenotypes from the Danish Medical Birth Register[34]: apgar5 (Apgar score, 5 min after birth), fvagt (birth weight in grams) and gest_age (gestational age in completed weeks). All used phenotypes, sample sizes and metadata are available at SD3-SD4.

### Multi-PGS models

We used the library of 937 PGS to train multi-phenotype predictors (multi-PGS) using two different algorithms (1) L1 penalized regression (lasso) as implemented in the R package glmnet[53] and (2) tree gradient boosting as implemented in the XGBoost (eXtreme Gradient Boosting) algorithm in the R package xgboost[21]. For both models, we trained a base model using only the covariates (sex, birth year and 20 PCs) and a full model using the covariates plus the 937 standarized LDpred2-auto PGS. For the base model, we used the glm function with the option family = "logistic". For lasso, we used the function cv.glmnet from the glmnet R package with the options alpha = 0 and family = "binomial" for binary phenotypes. The covariates were not regularized by giving them a penalty factor of 0 with the option penalty.factor, while the rest of the PGS were given a penalty factor of 1. For XGBoost, we used the xgboost function from the xgboost R package with options eta = 0.01 and nrounds = 10. We used objective = "binary:logistic" for binary phenotypes and objective = "reg:squarederror" for continuous variables.

### BLUP PGS

We computed an internal best linear unbiased prediction (BLUP) PGS trained on the individual-level data. We obtained the per-SNP prediction betas with BOLT-LMM[54,55] (using the flag –predBetasFile) on the

set of 1,118,443 HM3 SNPs on iPSYCH. Depending on the polygenicity of the phenotype, BOLT-LMM computes a mixture-of-Gaussians prior or a single-Gaussian BOLT-LMM-inf model, equivalent to best linear unbiased prediction (BLUP). In the case of psychiatric disorders, our results show that BOLT-LMM-inf is always the model selected and therefore we refer to the BOLT-LMM PGS as BLUP PGS.

## Evaluation of prediction accuracy

For each phenotype, we used a fivefold cross-validation scheme to obtain out-of-sample prediction accuracy estimates. The prediction was evaluated by (1) adjusted variance explained in the liability scale. We used population prevalences specified in SD2 to convert the variance explained in a linear regression to the liability scale[56]. The adjusted $R^2$ was defined as the variance explained by the full model after accounting for the variance explained by the base model as R2_adjusted = (R2_full - R2_cov)/(1 − R2_cov). (2) Odds ratio (OR) of the 5th quintile to the middle quintile. All OR were calculated from a logistic regression model based on the PGS percentiles, sex, birth year and first 20PCs.

## Reporting summary

Further information on research design is available in the Nature Portfolio Reporting Summary linked to this article.

## Data availability

The multi-PGS lasso weights generated in this study have been deposited in the figshare database (https://doi.org/10.6084/m9.figshare.23597019.v1). The iPSYCH and Danish ANGI data are available under restricted access as the data are protected by Danish legislation, access can be obtained after approval by the iPSYCH Data Access Committee and can only be accessed on the secured Danish server GenomeDK (https://genome.au.dk). For data access and correspondence, please contact C.A. (albinanaclara@gmail.com) or B.J.V. (bjv.ncrr@au.dk). The PGS library metadata generated in this study is provided in the Supplementary Information/Source Data file. The GWAS summary statistics data used in this study are available in the GWAS Catalog database (https://www.ebi.ac.uk/gwas/, downloaded on 09/09/2020), GWAS Atlas UKB2 data freeze v20191115 (https://atlas.ctglab.nl/) and PGC downloads (https://www.med.unc.edu/pgc/download-results/).

## Code availability

All code used in this project is available in GitHub https://github.com/ClaraAlbi/paper_multiPGS[57].

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

## Acknowledgements

C.A., B.J.V. and F.P. were supported by the Danish National Research Foundation (Niels Bohr Professorship to Prof. John McGrath), the Lundbeck Foundation Initiative for Integrative Psychiatric Research, iPSYCH (R102-A9118, R155-2014-1724 and R248-2017-2003), and a Lundbeck Foundation Fellowship (R335-2019-2339). C.A. was supported by a Willam Demant Fonden fellowship. I.B. was also supported by the Swedish Brain Foundation and Fredrik och Ingrid Thurings Stiftelse. AN data are from the Anorexia Nervosa Genetics Initiative, an initiative of the Klarman Family Foundation, and extendented with support from the Lundbeck foundation (R276-2017-4581). High-performance computer capacity for handling and statistical analysis of iPSYCH data on the GenomeDK HPC facility was provided by the Center for Genomics and Personalized Medicine and the Centre for Integrative Sequencing, iSEQ, Aarhus University, Denmark (grant to A.D.B.).

## Author contributions

C.A. performed the analyses. C.M.B., L.V.P., E.A., J.G., M.N., D.M.H., T.W., A.D.B., P.B.M., J.J.M., B.M.N. and F.P. performed sample and/or data provision and processing. C.A. and B.J.V. wrote the manuscript. C.A., Z.Z., A.J.S., A.I., H.A., I.B., E.A., J.G., J.J.M., B.M.N., F.P. and B.J.V performed core revision of the manuscript. B.J.V and F.P. supervised the study. All authors contributed to critical revision of the manuscript.

## Competing interests

C.M.B. reports: Lundbeckfonden (grant recipient); Pearson (author, royalty recipient); Equip Health Inc. (Stakeholder Advisory Board). B.M.N. is a member of the scientific advisory board at Deep Genomics and Neumora, consultant of the scientific advisory board for Camp4 Therapeutics and consultant for Merck. B.J.V. is on Allelica's international advisory board. The remaining authors declare no competing interests.
