## [Peer Review File · Nature Communications]

Multi-PGS enhances polygenic prediction by combining 937 polygenic scoresREVIEWER COMMENTS

Reviewer #1 (Remarks to the Author):

The paper "Multi-PGS enhances polygenic prediction: weighting 937 polygenic scores" presents a novel approach to using polygenic scores (PGSs) for predicting traits by combining almost a 1000 PRSs using lasso regularization to improve prediction accuracy. This approach is demonstrated to have potential benefits compared to using a single PRS and has applications for predicting traits without large GWAS sumstats available.

I found this work to be very well-written and informative. The idea of using multiple PRSs to improve prediction accuracy has been an approach that I have wanted to try for a long time. Organizing the 1000 PGS the way they describe is a large task and the results of the study are convincing that this is a good approach. The application to predicting traits without large GWAS is particularly useful, as it opens up new avenues for research and discovery. Overall, I believe this paper makes a valuable contribution to the field and I would highly recommend it to others interested in the use of PRSs for trait prediction. I do have a few minor suggestions and corrections for the work:

- I don't find Figure 2B very helpful. The lines are mostly overlapped and I don't see 95% CIs there to really distinguish each OR from each other. One alternative is just to show just the highest quintile compared to the middle quintile for each of the four models. Then we can see the difference in the ORs for just this comparison. Maybe other quintile comparisons in Supplemental? It would also be helpful to include a brief description of each Risk Score in the legend particularly the lassoPGS_xgboostCOV one.
- The same suggestion can be said for Figure 4B as what I stated above.
- The description of the results for Figure 2 is excellent and the conclusions to use lassoPGS are reasonable.
- I notice that for AFF analyses, both MDD Howard and MDD PGC2 are in the top 3 PRS. This seems to imply that each PRS is capturing something in the AFF diagnosis in iPSYCH. This is potentially important to point out because this makes me think that we should include all possible GWAS sumstats (with adequate training sample size) for a trait rather than just choosing the one with the largest sample size (and most heterogeneity).
- I also think its worth pointing out that this approach could be useful for subphenotype analysis within disorders because there are no published GWAS sumstats of subphenotypes (e.g. psychosis within bipolar disorder).
- I clicked on the github links for the Supp Tables and none of them work. I was able to navigate to the github instead and find them. It's also
- Y-axis label on SF5 should be "Mean adjusted AUC"
- I'm not sure why the ADHD PRS is not included in Figure 3 for ADHD. It is above chronic pain and BMI PRSs in terms of the lasso weight in SF6.

- I don't really get some of the paired PRSs selected for Figure 5. For example, F10-F19 is paired with PGC_CUD but why not PGC_AUD or MVP_OUD? It's not really surprising that the lasso did better in this case. Because of this, I don't know if there is much point to include a singleGWAS comparison in these cases and I think I would prefer to only see the lassoPGS based on what has been shown previous to this Figure that lassoPGS is better than a singlePGS. And to interpret the X-fold increase in performance seems misleading.

Reviewer #2 (Remarks to the Author):

This manuscript reports on an interesting approach to using information from GWAS for a large number of phenotypes to develop polygenic models for six psychiatric disorders. Of particular note is that the method is agnostic with respect to any genetic correlation between the phenotypes used to develop the model and the phenotype of interest. Fundamentally the methods appear to be sound and the results are interesting.

However, I found this manuscript very difficult to follow and had to read and re-read some of the sections multiple times in order to understand what the authors had done and thus to interpret the results. I did not find figure 1 very helpful in understanding each step of the process. I think one reason for this is the use of the term PGS (polygenic score) in multiple contexts. It might help to distinguish a polygenic model, which is a set of variants and their weights, from a polygenic score which is the result of application of a polygenic model to an individual's genotypes, or to come up with some additional terms to refer to different elements that underpin the method. The introduction and the methods elements of the results section could be substantially improved (particularly for the non-specialist reader).

Just one example is the sentence: "Next, we investigated which PGS in the multi-PGS model were the ones contributing the most to increasing prediction accuracy. The number of non-zero PGS in each multi-PGS model ranged from 10 to 154, where the number of PGS included correlated with the number of samples in the training set."

I have written the following to summarise my understanding of the methods. I think that a similar style would help throughout. "We used publicly available summary statistics for 937 different phenotypes to develop 937 polygenic models (PGM) using LDpred-auto. These models were then applied to the genotypes of individuals in the Lundbeck Foundation Initiative for Integrative Psychiatric Research dataset so that each individual had 937 polygenic scores (PGS). These individual level PGS were then used to develop prediction models (multiPGM) for each of six psychiatric disorders using both a linear model (lasso penalized regression) and a non-linear model (scalable gradient boosted trees). The multiPGMs for each phenotype comprises a set of weights for each of the 937 polygenic scores which are applied to each individual to derive the multi-polygenic score. We then compared the performance

of the multiPGM with a standard PGM derived from the summary statistics from the largest publicly-available GWAS for each disease phenotype.”

P5, para 1. It is not clear to me how “comparing the performance of linear models (lasso penalized regression; multiPGS_lasso) and non-linear models (boosted gradient trees: multiPGS_XGBoost) to predict the 6 major psychiatric disorders” enables one to study “the relationship between the covariates (sex, age and first 20 PCs) and the 937 PGS”.

P9, para 2. If I have understood correctly the authors have compared a PGM developed using the publicly-available summary statistics from an independent GWAS with a PGM developed using the individual level genotypes from the iPSYCH dataset (BLUP method). Given that the sample size of the external GWAS and iPSYCH are different (as the authors note) the comparison seem uninformative. Furthermore, as the BLUP PGM was fit using the data used for the comparison there is some over-fitting, which is not a problem for the external GWAS.

In this paragraph the authors state that they compare the PGM models described above with “re-weighting the set of PGS, which uses both types of data”. It is unclear what the ‘set of PGS’ is (?the 937 PGM) and what is meant by reweighting. Nor can I understand what the result of this comparison was. This paragraph has a structure in which it is stated “First, ...”, but then there is no follow-up ‘second’.

Supp figure 2. The axes should not be labelled with abbreviations. The meanings of the standard deviation of the genotyped/imputed data and the standard deviation GWAS summary statistics are not clear to me.

Thank you for giving us the opportunity to revise and resubmit our manuscript. We believe that our revised manuscript has improved significantly based on the suggested revisions. Below is a point-by-point response to all the comments provided by the reviewers. Our responses are written in blue, the citations from the (new) manuscript text are in green, the original comments are in black.

Reviewer #1 (Remarks to the Author):

The paper "Multi-PGS enhances polygenic prediction: weighting 937 polygenic scores" presents a novel approach to using polygenic scores (PGSs) for predicting traits by combining almost a 1000 PRSs using lasso regularization to improve prediction accuracy. This approach is demonstrated to have potential benefits compared to using a single PRS and has applications for predicting traits without large GWAS sumstats available.

I found this work to be very well-written and informative. The idea of using multiple PRSs to improve prediction accuracy has been an approach that I have wanted to try for a long time. Organizing the 1000 PGS the way they describe is a large task and the results of the study are convincing that this is a good approach. The application to predicting traits without large GWAS is particularly useful, as it opens up new avenues for research and discovery. Overall, I believe this paper makes a valuable contribution to the field and I would highly recommend it to others interested in the use of PRSs for trait prediction. I do have a few minor suggestions and corrections for the work:

We thank the reviewer for this positive assessment and that they find our work to be both well-written and informative.

- I don't find Figure 2B very helpful. The lines are mostly overlapped and I don't see 95% CIs there to really distinguish each OR from each other. One alternative is just to show just the highest quintile compared to the middle quintile for each of the four models. Then we can see the difference in the ORs for just this comparison. Maybe other quintile comparisons in Supplemental? It would also be helpful to include a brief description of each Risk Score in the legend particularly the lassoPGS_xgboostCOV one.

We agree that figures 2B and 4B Figures could be improved. We have now followed the reviewer's suggestion and simplified the comparison to only show the risk score quintiles Q3 vs Q5. As an example Figure 2B now looks like this:

Moreover, we have added supplementary figures with the full OR comparisons which includes all risk score quintile OR compared to the middle for both Figure 2 and Figure 4 (SF6 and SF14). On the second issue raised for Figure 2B, we have now added extra text in the legend that reads “The models MultiPGS_lasso and MultiPGS_xgboost were generated with lasso regression and XGBoost respectively, using the 937 PGS and the covariates as explanatory variables. The model MultiPGS_lassoPGS_xgboostCOV was generated with lasso regression, combining the 937 PGS and the predicted values of an XGBoost model that included only the covariates.”.

- The same suggestion can be said for Figure 4B as what I stated above.

We followed the same procedure as above to modify Figure 4B (and SF14).

- The description of the results for Figure 2 is excellent and the conclusions to use lassoPGS are reasonable.

We are glad that the reviewer appreciates these findings.

- I notice that for AFF analyses, both MDD Howard and MDD PGC2 are in the top 3 PRS. This seems to imply that each PRS is capturing something in the AFF diagnosis in iPSYCH. This is potentially important to point out because this makes me think that we should include all possible GWAS sumstats (with adequate training sample size) for a trait rather than just choosing the one with the largest sample size (and most heterogeneity).

This is an excellent point that we discussed internally but did not include much text about in the manuscript. We now include the following text associated with Figure 3, which hopefully addresses this issue and also a comment from Reviewer 2.

“Among the top 5-weighted PGS for each psychiatric disorder, we identified the PGS from the PGC GWAS excluding iPSYCH samples for all disorders. For affective disorder, three depression PGS were selected, two from the PGC (Wray et al. 2018); (Howard et al. 2019) (tagged MDD-PGC2 and MDD Howard) and one from a UK Biobank GWAS on depressive symptoms (Baselmans et al. 2019) (tagged Depressive symptoms). These results suggest that non-overlapping signals from multiple GWAS of similar phenotypes can be combined to increase overall prediction accuracy. Interestingly, the PGS from the PGC ADHD GWAS excluding iPSYCH samples ranked 5th for ADHD. This study had only 4,225 cases excluding iPSYCH cases, similarly to the ASD study, with 5,305 cases excluding iPSYCH.”

- I also think it's worth pointing out that this approach could be useful for subphenotype analysis within disorders because there are no published GWAS sumstats of subphenotypes (e.g. psychosis within bipolar disorder).

We appreciate that the reviewer also sees potential in broader applications of our method. The case with psychosis within bipolar disorder (BD) is a perfect example of one of the applications we wanted to highlight. Although we showcased this with the autism spectrum disorder subphenotypes, we now include some text about psychosis and BD in the discussion.

“The multiPGS predictors do not require PGS for the target phenotype of interest to be available in the PGS library used. This application is particularly interesting for sub-phenotype analyses within diseases, where GWAS summary statistics are not generally available for the sub-phenotypes. We demonstrated in practice how these multi-PGS could be generated for various psychiatric sub-diagnoses e.g. different ICD10 subcodes within Autism Spectrum Disorder (ICD10 F8). Similar multi-PGS method could also be applied to other sub-phenotypes of psychiatric disorders like psychosis within bipolar disorder, as defined in Hasseris et al. 2023. Another exciting application we explored is the case-case prediction, where multi-PGS models can be trained for highly comorbid disorders. In this last category, we highlight the relatively high prediction accuracy of our predictor of ADHD cases from a pool of ADHD-ASD cases.”

Unfortunately, we were not able to generate psychosis-related multi-PGS due to current data availability. However, we plan to use the polarity definitions proposed by Hasseris et al 2023 (Hasseris et al. 2023) in future work.

- I clicked on the github links for the Supp Tables and none of them work. I was able to navigate to the github instead and find them. It's also

We apologize for the link issue. We have tried the links in different scenarios and still work for us, but in case there are further issues we now also provide the datasets as attached .csv files.

- Y-axis label on SF5 should be “Mean adjusted AUC”
Fixed (see below). In the case of the AUC measure, it is not “adjusted” as the models did not include the covariate adjustment, only PGS.

- I'm not sure why the ADHD PRS is not included in Figure 3 for ADHD. It is above chronic pain and BMI PRSs in terms of the lasso weight in SF6.

We thank the reviewer very much for spotting this error. The PGSs in Figure 3 were incorrectly selected by their individual adjusted R2 instead of by their lasso weights, and thus the discordance with SF7. Please find below the modified Figure 3 where the top 5 lasso weighted PGS are presented, ordered by the adjusted R2 for comparison reasons. We are interested in presenting the results based on the lasso weight ranking because

this weight highlights the feature selection quality of the lasso model. We note that the ranking based on R2 is not equivalent to the ranking based on lasso weights.

For example, in the particular case of predicting ADHD, the external ADHD PGS explains less variance in terms of R2 alone than the multisite chronic pain PGS (1% vs. 1.3% respectively) but the lasso weight for the ADHD PGS is larger than for the multisite chronic pain PGS (0.76 vs. 0.75 respectively). Because the variance in the outcomes explained by some of these PGS is overlapping, the lasso weights are adjusted accordingly in the model to get rid of the redundancy. Nevertheless, we consider these differences to be minimal.

We also modified SF13, as it had the same original problem as Figure 3.

ADHD

AFF

AN

ASD

BD

SCZ

- I don't really get some of the paired PRSs selected for Figure 5. For example, F10-F19 is paired with PGC_CUD but why not PGC_AUD or MVP_OUD? It's not really surprising that the lasso did better in this case. Because of this, I don't know if there is much point to include a singleGWAS comparison in these cases and I think I would prefer to only see the lassoPGS based on what has been shown previous to this Figure that lassoPGS is better than a singlePGS. And to interpret the X-fold increase in performance seems misleading.

We thank the reviewer for raising this point, it is true that some of the comparisons were chosen just to have something to compare to because the "best" option maybe wasn't available. We still believe it is useful for the reader to be able to see a comparison of multi-PGS vs. single-PGS but agree that Figure 5 may not have provided a fully fair comparison. To remedy this, we have now created a new Figure 5 where the comparison of the multi-PGS is against the PGS with the largest weight in the lasso model. This PGS is now labeled in the Figure to minimize confusion of what is being compared. The new Figure 5 now aligns better with Figure 4, where top 5 single PGS with the largest lasso weights are also shown.

We have also restrained from using X-fold comparisons in this context, we agree with the reviewer that the pairs of multi-PGS - single-PGS are not the most fair comparison as they are generally not for the same phenotype (also removed from the Abstract). We have nevertheless retained the variance explained.

Reviewer #2 (Remarks to the Author):

This manuscript reports on an interesting approach to using information from GWAS for a large number of phenotypes to develop polygenic models for six psychiatric disorders. Of particular note is that the method is agnostic with respect to any genetic correlation between the phenotypes used to develop the model and the phenotype of interest. Fundamentally the methods appear to be sound and the results are interesting.

However, I found this manuscript very difficult to follow and had to read and re-read some of the sections multiple times in order to understand what the authors had done and thus to interpret the results. I did not find figure 1 very helpful in understanding each step of the process. I think one reason for this is the use of the term PGS (polygenic score) in multiple contexts. It might help to distinguish a polygenic model, which is a set of variants and their weights, from a polygenic score which is the result of application of a polygenic model to an individual's genotypes, or to come up with some additional terms to refer to different elements that underpin the method.

The introduction and the methods elements of the results section could be substantially improved (particularly for the non-specialist reader).

Just one example is the sentence: "Next, we investigated which PGS in the multi-PGS model were the ones contributing the most to increasing prediction accuracy. The number of non-zero PGS in each multi-PGS model ranged from 10 to 154, where the number of PGS included correlated with the number of samples in the training set."

I have written the following to summarise my understanding of the methods. I think that a similar style would help throughout. "We used publicly available summary statistics for 937 different phenotypes to develop 937 polygenic models (PGM) using LDpred-auto. These models were then applied to the genotypes of individuals in the Lundbeck Foundation Initiative for Integrative Psychiatric Research dataset so that each individual had 937 polygenic scores (PGS). These individual level PGS were then used to develop prediction models (multiPGM) for each of six psychiatric disorders using both a linear model (lasso penalized regression) and a non-linear model (scalable gradient boosted trees). The multiPGMs for each phenotype comprises a set of weights for each of the 937 polygenic scores which are applied to each individual to derive the multi-polygenic score. We then compared the performance of the multiPGM with a standard PGM derived from the summary statistics from the largest publicly-available GWAS for each disease phenotype."

We thank the reviewer for considering our method and results interesting. We have tried throughout the revised version of our manuscript to clarify the methodology for the non-

specialist reader. We agree that our notation can be confusing, and we have now tried to clarify what we mean by these terms in the text. However, we would prefer using PGS effect sizes instead of PGM. The reason is that in our previous publications we have generally referred to these as PGS or PRS, and would prefer continuing doing so to be consistent. We use as example the PGS Catalog, which actually provides PGS effect sizes, rather than actual individual scores. We believe it is a common terminology to talk about both, because it is straightforward to go from effect sizes (for each genetic variant) to scores (for each individual). We have however tried to improve the overall clarity of the manuscript and be more specific about what we mean when referring to PGS.

The following text, partially based on the input from the reviewer, is now included in the subsection “Overview of methods” within Results, together with a revised, hopefully clearer version of Figure 1:

“Here we summarize the framework used for generating the proposed multi-PGS. This framework consists of three steps: **Step 1** - Build PGS Library, **Step 2** - Train Multi-PGS Models and **Step 3** - Evaluate models (Figure 1). In Step 1, a large, agnostic library of PGS is generated by running LDpred2-auto¹⁶ on publicly available GWAS summary statistics (GWAS Catalog²⁶, GWAS ATLAS²⁷, PGC²⁸ etc.). In Step 2, the PGS library is standardized (i.e., mean 0 and variance 1) and used to develop prediction models (multi-PGS) for a target outcome using both a linear model (lasso penalized regression) and a non-linear model (boosted gradient trees, XGBoost). The multi-PGS models include sex, age and 20 first PCs as covariates. Finally in Step 3, the prediction accuracy of the multi-PGS is evaluated and benchmarked against the prediction accuracy of single PGS and another multivariate PGS method, wMT-SBLUP⁸. We used 5-fold cross-validation to alternate between Step 2 and Step 3 to get out-of-sample prediction accuracy estimates.”

Using the proposed multi-PGS framework, we generated a library of 937 PGS (described in detail in Supplementary Text) and projected it into the genotypes of individuals in iPSYCH. We then trained multi-PGS models for 6 major psychiatric disorders: attention-deficit/hyperactivity disorder (ADHD), affective disorder (AFF), anorexia nervosa (AN), autism spectrum disorder (ASD), bipolar disorder (BD) and schizophrenia (SCZ). We focus the first part of the results section on these 6 psychiatric disorders and extend the multi-PGS application to other 62 ICD10 code disease definitions, continuous phenotypes and case-case classification in the last result section.”

Figure 1: Overview of the multi-PGS framework.

The way the multi-PGS framework is introduced in the Main section has also been updated with the Step by Step notation.

The whole first paragraph on “Comparison between single PGS and multi-PGS predictors” describing Figure 3 has now been re-written to be more detailed (See comment to Reviewer 1). We hope the reviewer finds the extended version improved.

P5, para 1. It is not clear to me how “comparing the performance of linear models (lasso penalized regression; multiPGS_lasso) and non-linear models (boosted gradient trees: multiPGS_XGBoost) to predict the 6 major psychiatric disorders” enables one to study “the relationship between the covariates (sex, age and first 20 PCs) and the 937 PGS”.

As previously observed for polygenic traits, genetic effects work in a mostly additive fashion (Palmer et al. 2023). Therefore, by combining the covariates together with the PGS the hypothesis was that a “boost” in prediction from the non-linear models would indicate non-linear interactions between e.g. sex and a/multiple PGS. Then, to further investigate if this was the case or if the boost in prediction was only due to nonlinear interactions exclusively from the covariates, we created the MultiPGS_lassoPGS_xgboostCOV model. Our results, in which we do not observe large non-linear interactions between covariates and PGS, and PGS optimally combined linearly, are consistent with those in the DeepNull models by the Google Health team (McCaw et al. 2022).

Still, we agree with the reviewer that that specific sentence can be confusing. We have now re-written this specific phrase to “We first studied risk prediction models that combine the covariates (sex, age and first 20 PCs) and the 937 PGS using linear models (lasso penalized regression; multiPGS_lasso) and non-linear models (boosted gradient trees: multiPGS_XGBoost) to predict ADHD, AFF, AN, ASD, BD and SCZ.”

Given the positive comments from Reviewer 1 on the section “Linear and non-linear combinations of PGS give comparable prediction results” we are reluctant to change much on this section. However, we have included extra text in the legend of Figure 2 to clarify the “mixed” multi-PGS construction:

“The models MultiPGS_lasso and MultiPGS_xgboost were generated with lasso regression and XGBoost respectively, using the 937 PGS and the covariates as explanatory variables. The model MultiPGS_lassoPGS_xgboostCOV was generated with lasso regression, combining the 937 PGS and the predicted values of an XGBoost model that included only the covariates.”

P9, para 2. If I have understood correctly the authors have compared a PGM developed using the publicly-available summary statistics from an independent GWAS with a PGM developed using the individual level genotypes from the iPSYCH dataset (BLUP method). Given that the sample size of the external GWAS and iPSYCH are different (as the authors note) the comparison seem uninformative. Furthermore, as the BLUP PGM was fit using the data used for the comparison there is some over-fitting, which is not a problem for the external GWAS.

We would like to thank the reviewer for allowing us to increase the interpretability of our results to the readers. In previous work, we compared the effect in prediction accuracy of training PGS on individual-level data vs. GWAS summary statistics, given the same sample size is available (Albiñana et al. 2021). The intention in this manuscript was not to compare the single PGS to the BLUP PGS, but to compare the multi-PGS to the BLUP PGS, as it uses the exact same individual-level sample for training.

It was not clear from the text that we have **also** used a 5-fold cross validation scheme to train the BLUP model, so that the PGS derived from this method is also not over-fitted. We have now added to the paragraph highlighted by the reviewer: “We used 5-fold cross validation for deriving both the multi-PGS and BLUP PGS, so that the reported adjusted R2 are out-of-sample estimates.”

In this paragraph the authors state that they compare the PGM models described above with “re-weighting the set of PGS, which uses both types of data”. It is unclear what the ‘set of PGS’ is (?the 937 PGM) and what is meant by reweighting. Nor can I understand

what the result of this comparison was. This paragraph has a structure in which it is stated “First, ...”, but then there is no follow-up ‘second’.

We strongly agree with the reviewer that the description of the results in the section “Combining hundreds of external PGS increases prediction over training only on the individual-level data” can be substantially improved, particularly for the non-expert reader. Following the intention to make our manuscript more comprehensible, we have completely re-written this section. Considering the reviewer’s comments, we have a more structured paragraph where the intentionality of the analyses and the description of the methods is more detailed.

In our original draft, we used the concept of re-weighting to refer to putting weights on individual PGS, which are already weighted allelic effects. Therefore **re**-weighting. We understand now that that terminology can be confusing and have rephrased all “re-weighting” to combining, including the title of the manuscript. We have also stopped using the word “set” in this context for similar reasons. We also hope that these changes help express the innovation of our results and why we then explored a broader application of these in the following section.

Supp figure 2. The axes should not be labelled with abbreviations. The meanings of the standard deviation of the genotyped/imputed data and the standard deviation GWAS summary statistics are not clear to me.

We thank the reviewer for pointing this out. The axes on SF2 have now been re-labelled and are more self-explanatory.

SF2. Example QC plot. The QC step is described in detail elsewhere (Privé et al. 2022). The trait shown is for PMID 30643258 (Karlsson Linnér et al. 2019) GWAS summary statistics (Automobile speeding propensity).

References

- Albiñana, Clara, Jakob Grove, John J. McGrath, Esben Agerbo, Naomi R. Wray, Cynthia M. Bulik, Merete Nordentoft, et al. 2021. "Leveraging Both Individual-Level Genetic Data and GWAS Summary Statistics Increases Polygenic Prediction." *American Journal of Human Genetics* 108 (6): 1001–11.
- Baselmans, Bart M. L., Rick Jansen, Hill F. Ip, Jenny van Dongen, Abdel Abdellaoui, Margot P. van de Weijer, Yanchun Bao, et al. 2019. "Multivariate Genome-Wide Analyses of the Well-Being Spectrum." *Nature Genetics* 51 (3): 445–51.
- Buniello, Annalisa, Jacqueline A. L. MacArthur, Maria Cerezo, Laura W. Harris, James Hayhurst, Cinzia Malangone, Aoife McMahon, et al. 2019. "The NHGRI-EBI GWAS Catalog of Published Genome-Wide Association Studies, Targeted Arrays and Summary Statistics 2019." *Nucleic Acids Research* 47 (D1): D1005–12.
- "Download Results – PGC." n.d. Accessed October 19, 2022. <https://pgc.unc.edu/for-researchers/download-results/>.
- Hasseris, Sofie, Clara Albiñana, Bjarni J. Vilhjalmsón, Preben B. Mortensen, Søren D. Østergaard, and Katherine L. Musliner. 2023. "Polygenic Risk and Episode Polarity Among

- Individuals With Bipolar Disorder.” *The American Journal of Psychiatry* 180 (3): 200–208.
- Howard, David M., Mark J. Adams, Toni-Kim Clarke, Jonathan D. Hafferty, Jude Gibson, Masoud Shirali, Jonathan R. I. Coleman, et al. 2019. “Genome-Wide Meta-Analysis of Depression Identifies 102 Independent Variants and Highlights the Importance of the Prefrontal Brain Regions.” *Nature Neuroscience* 22 (3): 343–52.
- Karlsson Linnér, Richard, Pietro Biroli, Edward Kong, S. Fleur W. Meddens, Robbee Wedow, Mark Alan Fontana, Maël Lebreton, et al. 2019. “Genome-Wide Association Analyses of Risk Tolerance and Risky Behaviors in over 1 Million Individuals Identify Hundreds of Loci and Shared Genetic Influences.” *Nature Genetics* 51 (2): 245–57.
- Maier, Robert M., Zhihong Zhu, Sang Hong Lee, Maciej Trzaskowski, Douglas M. Ruderfer, Eli A. Stahl, Stephan Ripke, et al. 2018. “Improving Genetic Prediction by Leveraging Genetic Correlations among Human Diseases and Traits.” *Nature Communications* 9 (1): 989.
- McCaw, Zachary R., Thomas Colthurst, Taedong Yun, Nicholas A. Furlotte, Andrew Carroll, Babak Alipanahi, Cory Y. McLean, and Farhad Hormozdiari. 2022. “DeepNull Models Non-Linear Covariate Effects to Improve Phenotypic Prediction and Association Power.” *Nature Communications* 13 (1): 241.
- Palmer, Duncan S., Wei Zhou, Liam Abbott, Emilie M. Wigdor, Nikolas Baya, Claire Churchhouse, Cotton Seed, et al. 2023. “Analysis of Genetic Dominance in the UK Biobank.” *Science* 379 (6639): 1341–48.
- Privé, Florian, Julyan Arbel, Hugues Aschard, and Bjarni J. Vilhjálmsson. 2022. “Identifying and Correcting for Misspecifications in GWAS Summary Statistics and Polygenic Scores.” *HGG Advances* 3 (4): 100136.
- Privé, Florian, Julyan Arbel, and Bjarni J. Vilhjálmsson. 2020. “LDpred2: Better, Faster, Stronger.” *Bioinformatics*, December. <https://doi.org/10.1093/bioinformatics/btaa1029>.
- Watanabe, Kyoko, Sven Stringer, Oleksandr Frei, Maša Umičević Mirkov, Christiaan de Leeuw, Tinca J. C. Polderman, Sophie van der Sluis, Ole A. Andreassen, Benjamin M. Neale, and Danielle Posthuma. 2019. “A Global Overview of Pleiotropy and Genetic Architecture in Complex Traits.” *Nature Genetics* 51 (9): 1339–48.
- Wray, Naomi R., Stephan Ripke, Manuel Mattheisen, Maciej Trzaskowski, Enda M. Byrne, Abdel Abdellaoui, Mark J. Adams, et al. 2018. “Genome-Wide Association Analyses Identify 44 Risk Variants and Refine the Genetic Architecture of Major Depression.” *Nature Genetics* 50 (5): 668–81.

REVIEWERS' COMMENTS

Reviewer #1 (Remarks to the Author):

The authors have sufficiently responded to all of my comments and I enjoy the changes made to satisfy the other reviewer in terms of clarity.

Reviewer #2 (Remarks to the Author):

All of the reviewers comments have been carefully addressed, except I still think the methods are somewhat arcane. The fact that the term PGS is widely used to mean a polygenic model (a set of SNPs and their weights) despite the term being an abbreviation of polygenic score - the application of a PGM to an individual set of genotypes to calculate a number - is irrelevant. If the common use of a term is inaccurate and causes confusion then the answer is to change common practice.

In the online methods it is stated that "Polygenic scores were derived using LDpred2-auto". I challenge the authors to derive a set of polygenic scores with a data set of genotypes and no other information using LDpred. LDpred generates a model, not a score. But with a specified polygenic model it would be possible to derive the polygenic scores using simple arithmetic.

Thank you for giving us the opportunity to revise and resubmit our manuscript. We believe that our revised manuscript has improved significantly based on the suggested revisions. Below is a point-by-point response to all the comments provided by the reviewers. Our responses are written in blue, the citations from the (new) manuscript text are in green, the original comments are in black.

REVIEWERS' COMMENTS

Reviewer #1 (Remarks to the Author):

The authors have sufficiently responded to all of my comments and I enjoy the changes made to satisfy the other reviewer in terms of clarity.

We would like to thank Dr. Coombes for taking their time to review our manuscript and for providing constructive feedback that has greatly improved our work.

Reviewer #2 (Remarks to the Author):

We would like to thank Prof. Pharoah for taking their time to review our manuscript and for providing constructive feedback that has greatly improved our work.

All of the reviewers comments have been carefully addressed, except I still think the methods are somewhat arcane. The fact that the term PGS is widely used to mean a polygenic model (a set of SNPs and their weights) despite the term being an abbreviation of polygenic score - the application of a PGM to an individual set of genotypes to calculate a number - is irrelevant. If the common use of a term is inaccurate and causes confusion then the answer is to change common practice.

We understand the concern on the ambiguity of using PGS and PGS model thorough the manuscript. To address this, we have now been very careful of not using the word PGS model when referring to prediction. Therefore, the use of the word **model** has been limited in the manuscript. We only use it when we are explicitly talking about training the lasso/XGBoost models (Step 2), but never once the weights from these models have been projected into the individuals. After that, we simply talk about the prediction accuracy of multi-PGS.

The following sentence has been added to a re-structured introduction section to reduce potential confusion “Multiple PGS and covariates can be combined using either a linear model (lasso penalized regression) or a nonlinear model (XGBoost) into a multi-PGS model. This model is then evaluated in an independent dataset in terms of the prediction accuracy of the multi-PGS.”

In the online methods it is stated that "Polygenic scores were derived using LDpred2-auto". I challenge the authors to derive a set of polygenic scores with a data set of genotypes and no other information using LDpred. LDpred generates a model, not a score. But with a specified polygenic model it would be possible to derive the polygenic scores using simple arithmetic.

We understand the reviewer's point here. Although this type of sentence is widespread in the literature, we have now modified to “Polygenic score weights were derived using LDpred2-auto” to be more specific on what was actually derived.